# A Multi-View Stereo Measurement System Based on a Laser Scanner for Fine Workpieces

**DOI:** 10.3390/s19020381

**Published:** 2019-01-18

**Authors:** Limei Song, Siyuan Sun, Yangang Yang, Xinjun Zhu, Qinghua Guo, Huaidong Yang

**Affiliations:** 1Key Laboratory of Advanced Electrical Engineering and Energy Technology, Tianjin Polytechnic University, TianJin 300387, China; songlimei@tjpu.edu.cn (L.S.); 1730041187@stu.tjpu.edu.cn (S.S.); xinjunzhu@tjpu.edu.cn (X.Z.); 2Tianjin University of Technology and Education, Tianjin 300222, China; yangyangang@tute.edu.cn; 3Department of Precision Instrument, Tsinghua University, Beijing 100084, China; yanghd@tsinghua.edu.cn

**Keywords:** multi-view 3D reconstruction, workpiece measurement, laser sensor, calibration, point clouds

## Abstract

A new solution to the high-quality 3D reverse modeling problem of complex surfaces for fine workpieces is presented using a laser line-scanning sensor. Due to registration errors, measurement errors, deformations, etc., a fast and accurate method is important in machine vision measurement. This paper builds a convenient and economic multi-view stereo (MVS) measurement system based on a linear stage and a rotary stage to reconstruct the measured object surface completely and accurately. In the proposed technique, the linear stage is used to generate the trigger signal and synchronize the laser sensor scanning; the rotary stage is used to rotate the object and obtain multi-view point cloud data, and then the multi-view point cloud data are registered and integrated into a 3D model. The measurement results show a measurement accuracy of 0.075 mm for a 360° reconstruction in 34 s, and some evaluation experiments were carried out to demonstrate the validity and practicability of the proposed technique.

## 1. Introduction

3D measurement finds wide applications in the fields of reverse engineering, game development, animation, antique digitization, etc. 3D laser scanning is a popular 3D measurement technology due to its flexibility and high accuracy [1,2,3,4]. Some techniques for 3D polyhedron reconstruction have been developed [5], but they still suffer from high measurement error. Compared with other non-contact measurement methods [6,7,8,9], laser triangulation has several advantages, including easy integration with computers, real-time processing, and high flexibility in 3D reconstruction. In contrast, it is difficult for structured light-based methods to achieve high measurement accuracy in measuring complex surfaces, and the confocal-based methods suffer from low sampling speed and efficiency in generating point cloud data. In this work, an efficient measurement system for small workpieces will be developed with a single line-scanning laser sensor.

In addition to materials and technology, a high machining precision usually is crucial to long-lasting service for fine machine parts or forged parts. There have been approaches developed for the digitization of 3D objects in reverse engineering using laser triangulation scanning or laser point cloud data [10,11]. Quite a lot of previous works [12,13,14,15] have been carried out on laser-scanning-based multi-view 3D measurement and reconstruction, where key issues including calibration and reconstruction were addressed. Regarding calibration, Zhao [16] put forward a ball-plate-based calibration method and proposed using the particle swarm optimization (PSO) algorithm to solve the critical point-cloud sphere center-fitting problem in the calibration. However, it is not suitable for on-line measurement due to its having too many iterations and initial conditions. A measurement system with a five-laser-beam probe was proposed by Lee and Shiou [17], where the calibration is performed based on the error patterns caused by object tilts, but the depth measuring range cannot be used for workpiece reconstruction. Dai et al. proposed a method to calibrate the revolution axis of a 360-deg based on minimizing the distance and the difference [18]. It can precisely acquire the axis’s position but has the limitation that the revolution axis should be close to parallel to the Y axis. Nguyen presented a new method for multiple-view shape and deformation measurement that combines the fringe projection and the digital image correlation (DIC) techniques [19]. The method is easy to extend to multiple sensors, but with uncertain accuracy for complex surfaces. Y. Li et al. presented a novel real-time reconstruction approach based on a 3D shape database [20]. However, the algorithm still needs manual intervention to identify the overlapped portions of multiple sections in the process of registration. Song proposed a new phase unwrapping algorithm based on three wavelength phase shift profilometry (TWPSP) to solve the problem of measurement speed and measurement range [21]. Although the method was more robust and accurate, the noise and matching error had some effect on the reconstruction results. He and Liang proposed the ICP (Iterative Closest Point) algorithm based on point cloud features (GF-ICP) to improve the convergence speed and the interval of convergence without setting a proper initial value [22]. Chen et al. developed a cost-efficient hand-held three-dimensional (3D) laser scanner for optical 3D laser scan data acquisition, and proposed an automatic registration algorithm-based ICP registration algorithm for partially overlapped point clouds [23]. However, it can be easily trapped into local minima, especially for data with high repetition or symmetry.

There are some issues with the above-mentioned systems. Some systems involve multiple sensors that are not portable, and significant reconstruction errors may be caused if the sensors are not well synchronized. In addition, compared with the traditional calibration method, calibration in point cloud coordinates can more directly reflect the 3D reconstruction accuracy within a proper measurement range.

In this work, a new multi-view high-precision 3D reconstruction system was developed to measure small workpieces. The system is based on a line-scanning laser to acquire the contour data of the object (axial section). The contours are transformed into point cloud data at each angle in the world coordinates, and finally a high-quality complete 3D point cloud is generated by translation scanning and rotating scanning. A calibration algorithm that uses gauge ladder and ball is developed and improved for the proposed system, and it can achieve high precision with respect to world coordinates and revolution axis calibration, leading to high-quality 3D point clouds during measurement.

The rest of the paper is organized as follows. In Section 2, the new measurement system is developed, and measurement errors are analyzed, which is helpful for the design of the calibration algorithm. In Section 3, the coordinate calibration method is developed based on the gauge ladder and turntable calibration is achieved by transforming the coordinate of every point cloud. In Section 4, the experiment results are analyzed using the standard CAD model and evaluated to demonstrate the effectiveness of the proposed system. Finally, some conclusions are drawn in Section 5.

## 2. Development of the MVS Measurement System

The schematic of the experimental setup for the 3D scanning system proposed in this paper is shown in Figure 1. It consisted of a LJ-V7060 profilometer, a lift stage, a rotary stage and an industrial computer. The laser sensor completes the image and profile acquisition during the measurement process, the lift stage and the rotary stage are used to control the attitude of the object, and the profile data acquired by the controller are used to data processing in computer.

The world coordinates system is based on the laser sensor, in which the x axis is parallel with the laser beam, the y axis is along the lift stage, and the z axis is pointing laser source. The workpiece is placed on the rotary stage for translation and rotation scans in turn. In addition, there are manual stages that can assist in accurate calibration.

As 3D point cloud data is used in the calibration and reconstruction methods in this paper, the model of the laser profiles is introduced according to the projection perspective transformation of the camera model by Wesley [24]. The transformation $T_{IW}$ of imaging coordinates (u,v) to world coordinates $\left(x_{w},y_{w},z_{w}\right)$ can established by,
(1)$$\{\begin{array}{c} x_{w}=\frac{\left(f\cdot t_{x}-X_{u}\cdot t_{z})(Y_{u}\cdot r_{8}-f\cdot r_{5})-(X_{u}\cdot r_{8}-f\cdot r_{2})(f\cdot t_{y}-Y_{u}\cdot t_{z}\right)}{\left(X_{u}\cdot r_{7}-f\cdot r_{1})(Y_{u}\cdot r_{8}-f\cdot r_{5})-(X_{u}\cdot r_{8}-f\cdot r_{2})(Y_{u}\cdot r_{7}-f\cdot r_{4}\right)}\\ \begin{array}{c} y_{w}=\frac{\left(f\cdot t_{y}-Y_{u}\cdot t_{z})(X_{u}\cdot r_{7}-f\cdot r_{1})-(Y_{u}\cdot r_{7}-f\cdot r_{4})(f\cdot t_{x}-X_{u}\cdot t_{z}\right)}{\left(Y_{u}\cdot r_{8}-f\cdot r_{5})(X_{u}\cdot r_{7}-f\cdot r_{1})-(Y_{u}\cdot r_{7}-f\cdot r_{4})(X_{u}\cdot r_{8}-f\cdot r_{2}\right)}\\ z_{w}=0 \end{array} \end{array}$$
where $\left(X_{u},Y_{u}\right)$ is the pixel corresponding to CCD/CMOS camera plane coordinates, $r_{1}\sim r_{9};t_{x};t_{y};t_{z}$ are transformation parameters of the coordinates, *f* is the effective focal length of the camera.

For any point *p* in the world coordinates, the transformation $T_{WI}$ of the world coordinates to imaging coordinates can be expressed as:(2)$$\{\begin{array}{c} X_{u}=f\cdot \frac{r_{1}x_{w}+r_{2}y_{w}+r_{3}z_{w}+t_{x}}{r_{7}x_{w}+r_{8}y_{w}+r_{9}z_{w}+t_{z}}\\ Y_{u}=f\cdot \frac{r_{4}x_{w}+r_{5}y_{w}+r_{6}z_{w}+t_{y}}{r_{7}x_{w}+r_{8}y_{w}+r_{9}z_{w}+t_{z}} \end{array}$$

The relationship between image coordinates and CCD coordinates is obtained by [24]:(3)$$\{\begin{array}{c} u=(s_{x}\cdot X_{d})/d_{x}+u_{0}\\ v=Y_{d}/d_{y}+v_{0} \end{array}$$
where $\left(X_{d},Y_{d}\right)$ is CCD coordinates after distortion correction and $s_{x};u_{0};v_{0}$ are camera internal parameters. Therefore, based on the solving of equations by the Cardano method [25], the transformation relationship between the real object and the laser strip can be established by:(4)$$\begin{array}{l} \left(x_{w},y_{w},z_{w})=T_{IW}(u,v\right)\\ \left(u,v)=T_{WI}(x_{w},y_{w},z_{w}\right) \end{array}$$

The multi-view 3D measurement process is shown in Figure 2. It can be seen that the system architecture is divided at four levels. The system integration level is responsible for ensuring the transmission of measurement data and synchronous sensors and motion devices. The data-collection level converts data to laser 3D point cloud by filtering and fitting. On the next level, various experiments are carried out that are designed to evaluate the measurement tools regarding calibration and transformation relations of the point clouds. Finally, through rough data registration by rigid transformation and fine registration by feature key-points, the multi-view reverse model was constructed completely for use in future works.

The configuration of our laser sensor is shown in Figure 3. It is comprised of a cylindrical objective lens, a semiconductor laser source, a GP-64 processor, a group of Ernostar camera lenses and a HSE3-CMOS imaging plane. The cylindrical objective lens, which consists of two semiconductor lasers, expands the laser beam into a stripe, allowing the laser to diffuse on the target and ensuring the power of the laser. In addition, then, the reflected laser stripes are refracted on HSE3-CMOS through the 2D lenses. Finally, the displacement and shape of the object are detected by the processor based on the position and shape calculation by means of the light stripes.

Compared with the traditional red laser, a blue laser with wavelength of 405 nm has a shorter wavelength, stronger anti-disturbance ability, and a more accurate stripe image in the imaging plane. At the same time, the sensor has the function of polarizing light, which can distinguish and eliminate the multi-reflection light, which hinders the measurement and improves measurement accuracy. HSE3-CMOS has a very high sensitivity and dynamic range. Even if the exposure time is only 15.6 microseconds, both the black surface with less reflected light and the smooth metal surface with intense reflected light can be accurately measured. The GP-64 processor, with its high-speed processing capability, can not only read CMOS shooting data and perform high-resolution sub-pixel processing, but can also perform high-precision linearization processing, data output, and so on, which greatly improves the sampling efficiency.

A synchronous acquisition method was designed for synchronous profile acquisition timing of the sensor and movement of the lift stage, fixing the y-axis coordinate unit without causing profile loss. The minimum trigger time $t_{up}$ is 500 ns, and the highest acquisition frequency can reach 8 kHz. There are many factors that affect the trigger signal, including excessive motor movement speed, undesirable encoder trigger mode, or too long a camera exposure. Figure 4 shows that the timing diagrams for image profile generation, along with more accurate trigger periods, also require calibration to determine the world coordinate system. The minimum scanning interval (y-axis resolution) after synchronization can reach 5 μm with a feeding speed of 200 mm/min.

According to the 3D measurement system established in this paper, the error sources can be divided into: (1) Scale caused by lens and scanning speed; (2) Skew caused by object placement and device installation; and (3) Rotation and translation due to turntable alignment.

Every camera exposure can obtain up to 640 pixels in the x axis, and the improved stripe center width is usually 3–4 pixels. Although the sensor uses a blue laser, which is more reliable, it will still be affected by ambient light, object color, and object material.

## 3. Coordinate Calculation and Calibration for the MVS Measuring System

### 3.1. Gauge Block Ladder Calibration

In general, measuring tools for visual measurements include gauge blocks, gauge balls, chessboards, etc. For the profilometer of this system, we designed a gauge block ladder method that can meet the precision and range requirements. The x-axis accuracy of the sensor is fixed, but in the y axis and z axis, the error needs to be analyzed and corrected. Gauge blocks are usually used to calibrate the precision of instrument and equipment systems in measurement. Figure 5 shows the variation tendency of 1.08 mm gauge block height error.

It’s obvious that the inclination in different directions will affect the measurement of errors. Therefore, the influence of the slant of the objective stage should be considered in the scanning process. The gauge block is scanned to record the data during the scanning process. Based on the measured variation tendency, the slope of transverse scanning is linear, and the maximum deviation is about 0.01 mm.

After median filtering, these profile data can be fitted by the least-squares method, and the methods can be implemented in MATLAB. The slant data in Figure 6 can be modified by the equation $y=1.167\times 10^{-5}x+1.078$ and $y=-1.048\times 10^{-5}x+1.085$. Therefore, the height difference between each gauge surface can be considered as the coordinate difference of the corresponding axis.

To verify the system’s axial measurability, an error analysis method for the gauge ladder is given in this experiment. Calibration by gauge ladders in laser sensors can generally correct the error in different scales. According to the data type and error type of the system, the gauge ladder point cloud data can be used to automatically measure the 3-dimensional error $e(e_{x},e_{y},e_{z})$ of the gauge and perform the compensation at the same time. The schematic of the automatic calibration method used in this paper is shown in Figure 7.

According to the point cloud data of the gauge ladder, we proposed a corresponding processing algorithm to automatically correct the coordinate axis. The steps of our gauge ladder data processing algorithm are as follows:Point cloud segmentation: The region growth method is used to segment the gauge block data $\left(P_{G1}\sim P_{Gn}\right)$ and filter external isolated points before saving in height order.Normal vector calculation: Based on the point cloud covariance, the average normal vector $\vec{n}$ of the whole data is calculated as the standard height difference direction of the gauge ladder.Centroid calculation: The centroid of each point cloud level is calculated, and the difference between the two adjacent centroid coordinates along the normal direction can serve as the measured value of the current level, while the error *e* is the difference between the standard value and the measured value.

Figure 8 shows the point cloud measurement results.

Once the error of each level has been calculated, a certain compensation ratio *k* is needed for each axial direction in the sensor, due to the scattering deviation of the laser. The compensation ratio *k* and minimal error $e_{\min}$ can be represented by:(5)$$\left|{\vec{e}}_{\min}\right|=\sum_{i=1}^{n}\left|\vec{k}M_{i}-S_{i}\right|$$

M,S are the measurement and standard values of the block ladders. To find the minimal error $e_{\min}$, the normalized k can be calculated as: (6)$$\vec{k}=\frac{\sum_{i=1}^{n}M_{i}S_{i}}{\sum_{i=1}^{n}M_{i}M_{i}}$$

Finally, the compensation ratio of each axis is set according to $k_{x},k_{y},k_{z}$. The point cloud processing and calibration is done using the Point Cloud Library (PCL) built-in functions [26].

### 3.2. Turntable Calibration

To determine the position of the turntable center axis in the world coordinates, it is necessary to calibrate the revolution axis and obtain the coordinates of a point on the axis in the world coordinates and the direction vector of the axis. The rotation angle of the turntable can be obtained by the encoder, while the position of the center axis of the turntable must be obtained by calibration parameters of the revolution axis. In this section, the gauge ball point cloud data on the basis of the laser sensor were used to calculate spherical center coordinates and establish a revolution axis coordinate system with the axis central coordinate and normal vector. Sitnik [27] proposed a calibration procedure with a flat plane to calculate the revolution axis, and Zheng [28] introduced a flexible new multi-view connection method with a calibration cylinder. Based on these two methods, we propose a calibration method suitable for the laser sensor, and calculate the revolution axis by spherical constraint and improved least-square method. Figure 9 shows the calibration sphere at different angles.

This method uses a point on the turntable axis and the direction vector of the axis to represent the position of the revolution axis in the world coordinates. The steps of our turntable calibration algorithm are as follows:At different positions of the rotating table, the 3D coordinate data $\left(x_{i}^{\left(n\right)},y_{i}^{\left(n\right)},z_{i}^{\left(n\right)}\right)$ of a partial sphere with radius $R_{B}$ are obtained by the sensor. These data are substituted into the space spherical equation, and the nonlinear equations are given by:(7)$$(x_{i}^{\left(n\right)}-x_{i}{)}^{2}+(y_{i}^{\left(n\right)}-y_{i}{)}^{2}+(z_{i}^{\left(n\right)}-z_{i}{)}^{2}=R_{B}^{2}$$By solving the equations, the spherical center coordinates $O_{n}(x_{n},y_{n},z_{n})$ of the spherical surface at this position can be calculated. Measured in the position *N*, the spherical center coordinate data of the group *N* were fitted together.The *N* group of spherical coordinate data is substituted into the space plane equation, and the N-dimensional linear equation system is constructed by:
(8)$$Ax_{n}+By_{n}+Cz_{n}+D=0\;(n=1,2\ldots N)$$Solving the equations by least-square method and the fitting plane equation, the equation $P_{B}$ can be considered the orbit equation of the center of a sphere. The normal vector of the plane *u* is:(9)$$u={\left[r_{1}\;r_{2}\;r_{3}\right]}^{\Gamma }={\left[\frac{A}{\sqrt{A^{2}+B^{2}+C^{2}}}\;\frac{B}{\sqrt{A^{2}+B^{2}+C^{2}}}\;\frac{C}{\sqrt{A^{2}+B^{2}+C^{2}}}\right]}^{\Gamma }$$The N group of spherical coordinate data $O_{n}(x_{n},y_{n},z_{n})$ were projected onto the calibrated rotation trajectory plane $P_{B}$, and then the coordinates of the corresponding projection points $O_{n}(x_{n}^{\prime },y_{n}^{\prime },z_{n}^{\prime })$ were obtained. The points found by the search algorithm that have the smallest distance from these projection points in the plane of rotation trajectory can be solved by: (10)$$f(x_{n}^{\prime },y_{n}^{\prime },z_{n}^{\prime })=\sum_{i=1}^{N}\left(\sqrt{{\left(x_{n}^{\prime }-x_{A}\right)}^{2}+{\left(y_{n}^{\prime }-y_{A}\right)}^{2}+{\left(z_{n}^{\prime }-z_{A}\right)}^{2}}-R_{p}\right)=\min$$
the point $O_{A}(x_{A},y_{A},z_{A})$ is the intersection of the revolution axis and the sphere center trajectory plane.

The method of measuring the 3D profile data of the standard ball based on translational scanning can be used to calibrate the turntable center axis by inverse planning of the spherical center coordinates for its simple calculation and easy manipulating.

Notably, in step 1, the Levenberg–Marquardt (LM) algorithm [29,30] is used to solve the non-linear least-squares problem of the spherical center and radius. According to the algorithm [29] and sphere equation, an objective function E,Φ can be established by:(11)$$\begin{array}{l} E={\sum_{i=1}^{n}\phi_{i}^{2}(c,r),\;\Phi =\left[\phi_{0},\phi_{1}\ldots \phi_{n}\right]}^{T}\\ \phi_{i}(c,r)={\left\|c-p_{i}\right\|}^{2}-r^{2}\;(i=1,2\ldots n) \end{array}$$

Where *c* is 3 × 1 sphere center; r is sphere radius; $p_{i}$ is 3 × 1 spherical data by scanning. In addition, an iterative relation is established based on 4 × n Jacobian Matrix *J*,
(12)$$J(c,r)=\frac{\partial \Phi }{\partial (c,r)}$$
(13)$$(J^{T}J+\lambda I)\delta =-J^{T}\Phi$$
where the damping factor λ is adjusted at each iteration, δ is a 4 × 1 vector, variables are updated with $c=c+\delta_{c},r=r+\delta_{r}$ in every iteration by solving δ until the change in δ is less than a certain threshold. According to the system error, δ can be set to 1e-4 in the program.

## 4. Experiments and Discussion

### 4.1. System Calibration Experiment

The proposed 3D point cloud calibration and alignment methods were incorporated into our multi-view stereo measurement system as shown in Figure 10. The motion system consists of a driving motor and a controller. Under the drive of the motor, the motion of 2-DOF can be completed. The lift stage (ETSS-200), with a resolution of 2.5 μm, is used to drive the sensor scanning; and the rotary stage (ERS-100) with a resolution of 0.00125° is fixed on the manual platform as a carrier.

As shown in Table 1, the laser sensor selected for the platform was of the new type LJ-V7060 profilometer (Keyence Co., Osaka, Japan).

To obtain the world coordinate system and improve the accuracy, gauge ladder and ball experiment results were obtained. Table 2 shows the error analysis of the gauge blocks ladder. In this experiment, three levels gauge blocks are selected to form the ladder. In addition, the regression height error of every ladder surface is then calculated, including mean error and root mean square (RMS).

According to the compensation ratio k calculated by the search method, the analysis shows that the maximum error of the system is 4.456 μm, the total mean error is 2.709 μm, and the total RMS is 3.720 μm, which is able to meet the measurement requirements of common workpieces. To validate the effectiveness of the corrected world coordinates, the diameter Mean Absolute Error (MAE) of a standard spherical surface data shows in Figure 11.

In the turntable calibration experiment, the sphere is sampled every 60°, the transformed and aligned spherical point cloud based on 6 views is shown in Figure 12.

The inverse calculation error of the spherical center can be used as the error evaluation standard of revolution axis calibration. This method is to calculate the spherical center coordinate error between the inverting transform from different angles of view to 0 degrees and the initial view from 0 degrees. According to calibration results, Table 3 shows the results of inverse calculation error by correction of the 6 angles of view.

In addition to the error analysis of the spherical center, it is necessary to compare the whole point cloud data with the standard model and further analyze the model error. A quadric surface was fitted to the MVS data of the standard sphere by Geomagic studio. The point cloud data from different angles of view are transformed into a world coordinate system according to the revolution axis, and then the model error is calculated by spherical fitting. Table 4 shows the error statistics of spherical surface and fitting result. The total point cloud filtering and more sophisticated registration are still needed for point redundancies and noises at edges and vertices.

### 4.2. 3D Reconstruction Experiment

The results of revolution axis calibration establish the rotation matrix $R_{u}$ and translation matrix $T(x_{A},y_{A},z_{A})$ of point clouds with different angles of view. In addition, the deviation of the spherical center coordinates can be corrected by $R_{u}$ and $T(x_{A},y_{A},z_{A})$, which are given by,
(14)$$T(x_{A},y_{A},z_{A})=\left[\begin{array}{cccc} 1 & 0 & 0 & -x_{A}\\ 0 & 1 & 0 & -y_{A}\\ 0 & 0 & 1 & -z_{A}\\ 0 & 0 & 0 & 1 \end{array}\right]$$
(15)$$R_{u}=\left[\begin{array}{cccc} r_{1}^{2}+(1-r_{1}^{2})\cos\theta & r_{1}r_{2}(1-\cos\theta )+r_{3}\sin\theta & r_{1}r_{3}(1-\cos\theta )+r_{2}\sin\theta & 0\\ r_{1}r_{2}(1-\cos\theta )-r_{3}\sin\theta & r_{2}^{2}+(1-r_{2}^{2})\cos\theta & r_{2}r_{3}(1-\cos\theta )-r_{1}\sin\theta & 0\\ r_{1}r_{3}(1-\cos\theta )-r_{2}\sin\theta & r_{2}r_{3}(1-\cos\theta )+r_{1}\sin\theta & r_{3}^{2}+(1-r_{3}^{2})\cos\theta & 0\\ 0 & 0 & 0 & 1 \end{array}\right]$$
where θ is rotation angle, and the corrected coordinates of point cloud $P^{\prime }$ can be transformed by,
(16)$$\left[\begin{array}{c} P^{\prime }\\ 1 \end{array}\right]=\left[\begin{array}{cc} R_{u} & T\\ 0^{T} & 1 \end{array}\right]\left[\begin{array}{c} P\\ 1 \end{array}\right]$$

Then, some preprocessing is needed to remove overlap and outlier points by PCL. Finally, the total point cloud is the 360-degree MVS model of the object. By using the above steps, a complete sphere can be obtained by the system. To examine the measurement uncertainty of the method, the spherical coordinate was computed from standard ball 3D data which were compared with the results of Geomagic. The statistical results of the two methods are shown in Table 5, and Figure 13 shows the RMSE of the reconstructed sphere 3D model.

During the research work in the previous sections, a part of a torx head bit workpiece on a screwdriver was able to be measured using the platform and method we proposed. Standard CAD models for such workpieces can be found online or from manufacturers. Figure 14 shows the workpiece surface profile information.

To verify the performance of the platform in real workpiece measurement, we evaluate the errors in different aspects by 3D comparison, as shown in Figure 15 and Figure 16. The total length of the model is 25 mm and the head diameter is 3.84 mm.

Figure 16 and Table 6 show that the new method we proposed achieve good results on 3D reconstruction accuracy and density. The registration accuracy reaches 0.075 mm, and because the sensor feeding speed during the measurement is F = 200 mm/min, then at least 4 views are required which in total requires about 34 s. A small portion of the object data was used to fix and clamp, which cannot displayed in the result.

## 5. Conclusions

This paper developed a multi-view stereo measurement system for fine workpieces based on a laser sensor. We established a platform that can realize automatic control of measurement and calibration processing with a lift stage and a rotary stage. In the course of profile extraction and gauge ladder calibration, the coordinates of points in a single view can be determined; during turntable calibration and rotation registration, a multi-view reverse engineering 3D model is described by point cloud data. Experiments demonstrate that precise full-view 3D models can be achieved with automatic scanning, which can meet the requirement of 3D detection, recognition and classification for various fine workpieces. In our future work, we will try to improve reconstruction quality by means of more refined registration and surface fitting methods. By increasing the sensor frame rate, the distortion will be reduced, because the sensor movement during the scan will decrease.

## Figures and Tables

**Figure 1 sensors-19-00381-f001:**
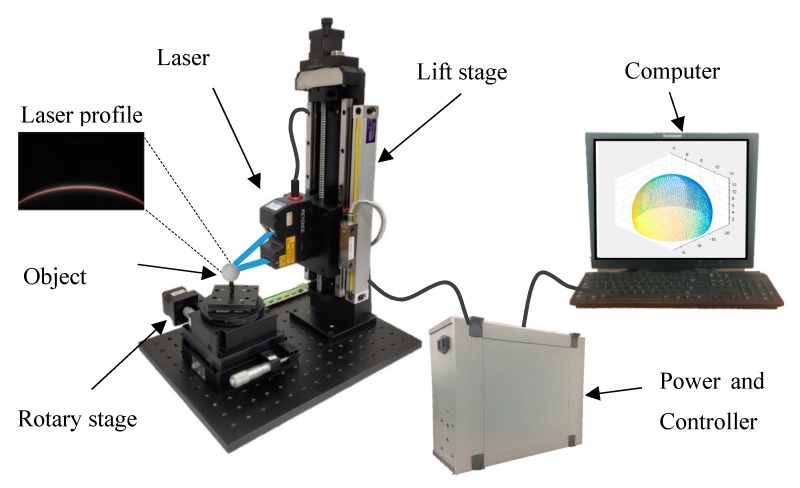
Schematic of the experimental setup for the 3D scanning system.

**Figure 2 sensors-19-00381-f002:**
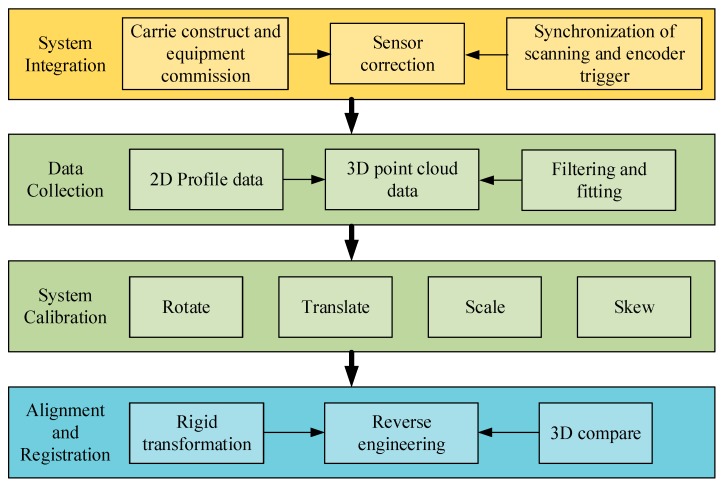
The block diagram of the multi-view 3D measurement process.

**Figure 3 sensors-19-00381-f003:**
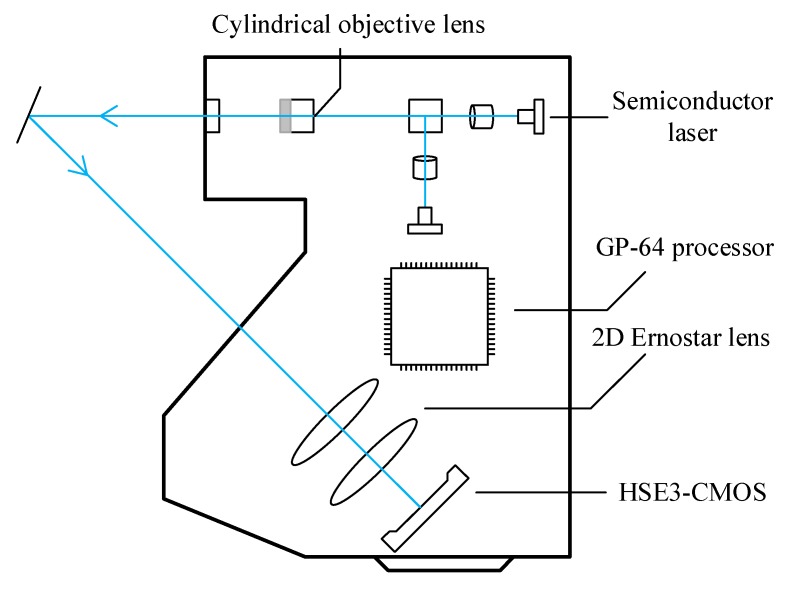
The configuration of our laser sensor.

**Figure 4 sensors-19-00381-f004:**
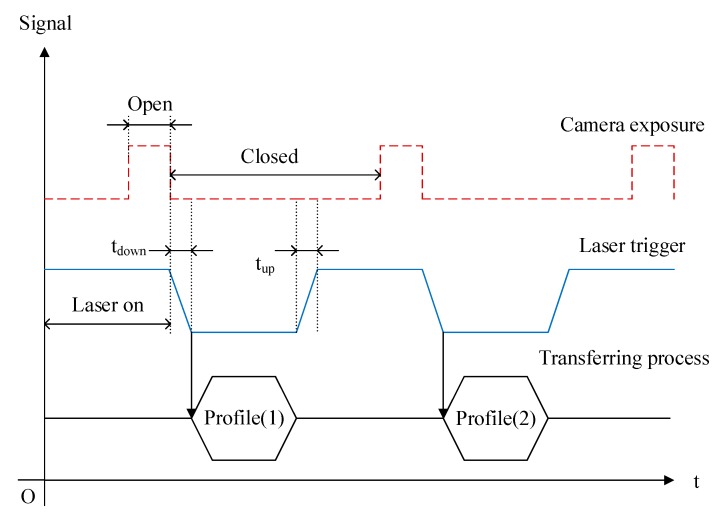
The timing diagrams for image profile generation.

**Figure 5 sensors-19-00381-f005:**
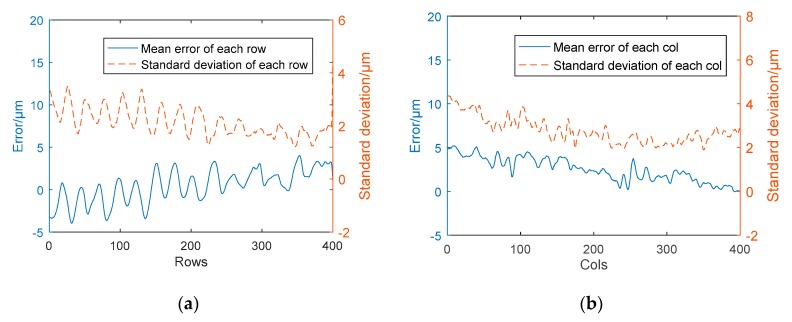
The variation tendency of 1.08 mm gauge block height error. (**a**) Measured error of each row; (**b**) Measured error of each col.

**Figure 6 sensors-19-00381-f006:**
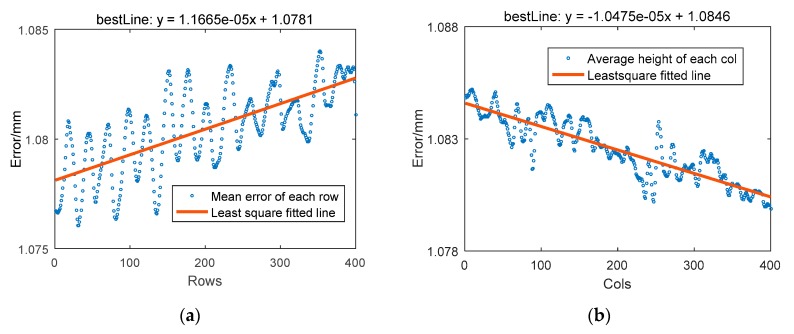
Slant correction based on the least-squares method. (**a**) Correction result of Figure 5a; (**b**) Correction result of Figure 5b.

**Figure 7 sensors-19-00381-f007:**
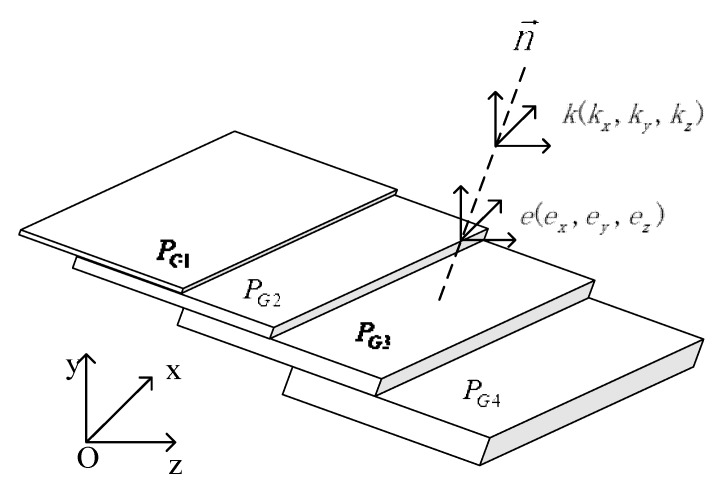
Schematic of the automatic calibration method.

**Figure 8 sensors-19-00381-f008:**
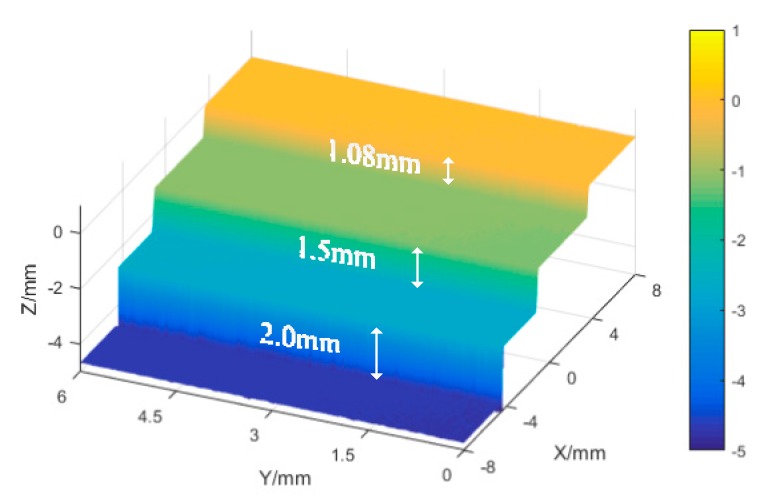
Point cloud measurement results of the gauge block ladder.

**Figure 9 sensors-19-00381-f009:**
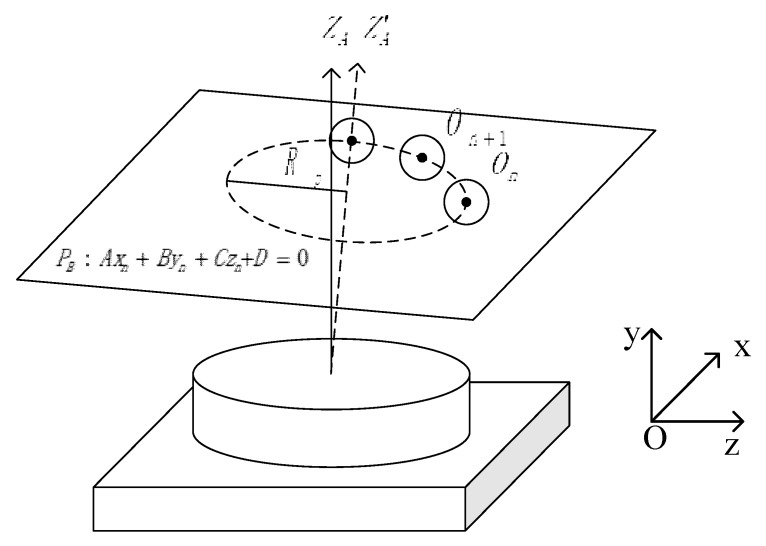
Rotation axis center calibration method.

**Figure 10 sensors-19-00381-f010:**
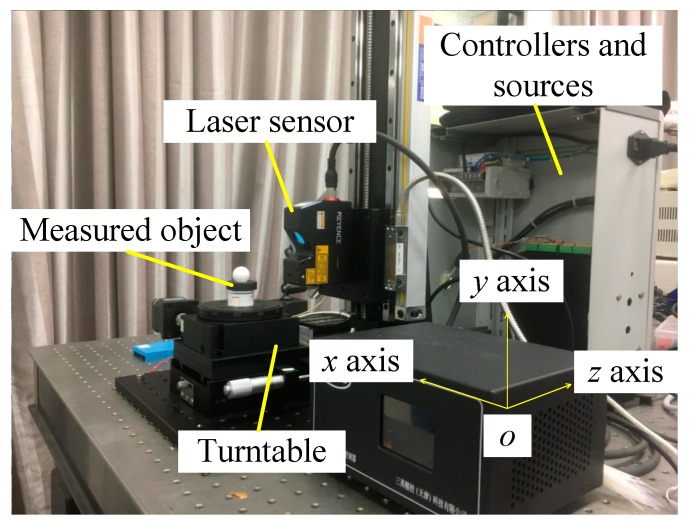
Components of the experiment platform and the world coordinate system used in this paper.

**Figure 11 sensors-19-00381-f011:**
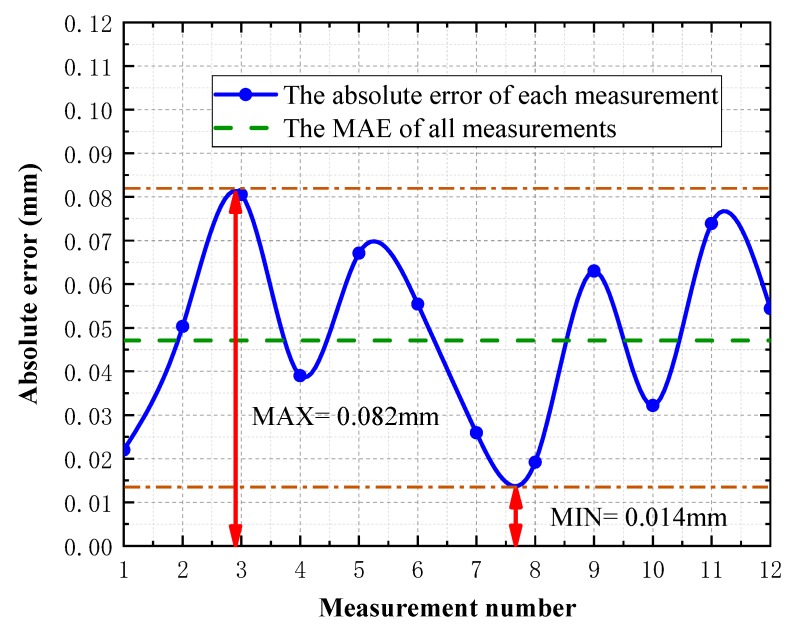
MAE of corrected spherical surface data diameter.

**Figure 12 sensors-19-00381-f012:**
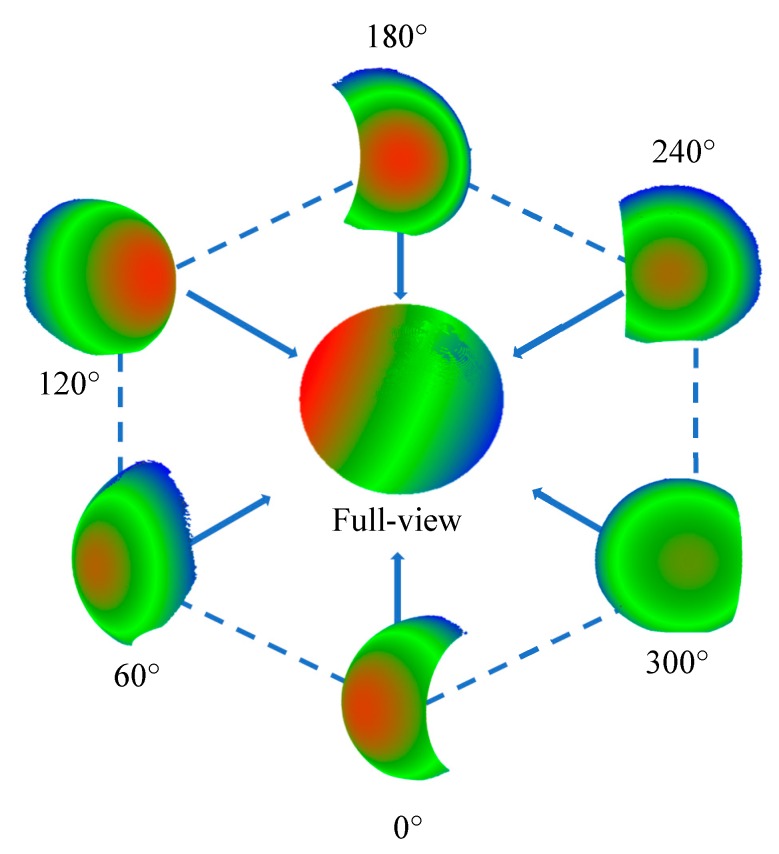
The transformed and aligned spherical point cloud based on 6 views. Because of the measurement range of the sensor, the complete spherical information cannot be scanned at once, and there will be a little loss at the top and bottom.

**Figure 13 sensors-19-00381-f013:**
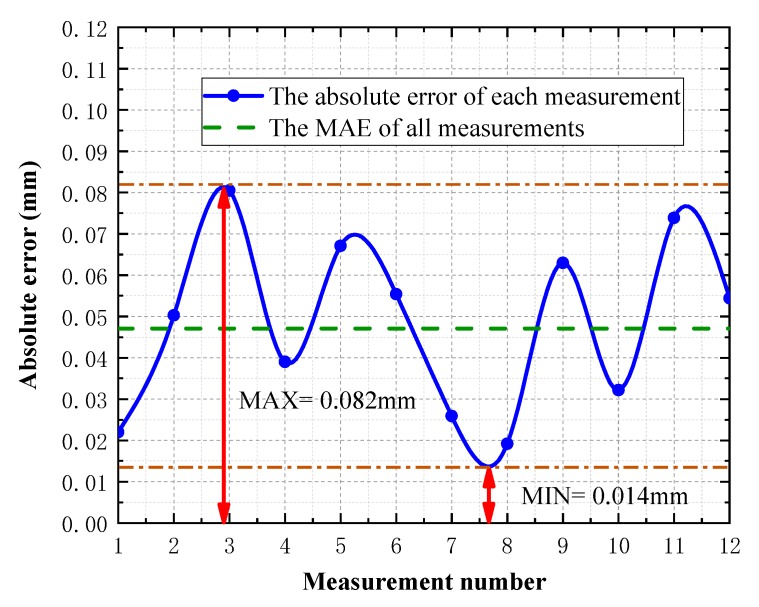
RMSE of the reconstructed sphere 3D model.

**Figure 14 sensors-19-00381-f014:**
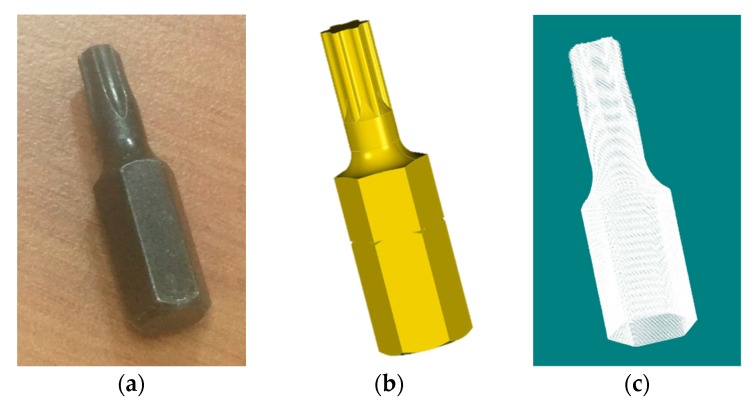
A workpiece surface profile information. (**a**) Physical photo of workpiece; (**b**) CAD model of workpiece; (**c**) MVS point cloud data of workpiece.

**Figure 15 sensors-19-00381-f015:**
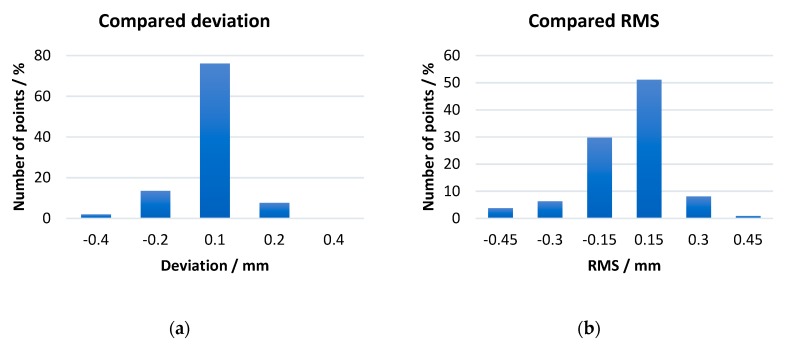
The best alignment result histograms. (**a**) Deviation distribution of alignment result; (**b**) RMS distribution of alignment result.

**Figure 16 sensors-19-00381-f016:**
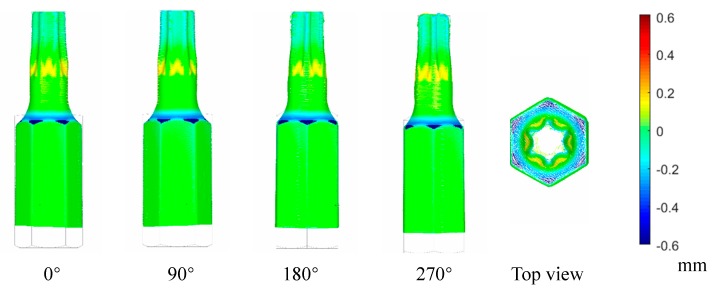
3D deviation distribution on multi-view workpiece surface.

**Table 1 sensors-19-00381-t001:** Technical features of the new LJ-V7060 profilometer.

Feature	Value
LED source (blue)	405 nm
Distance measurement	60 mm ± 8 mm
Spot size	45 μm
Laser output	10 mW
Sampling cycle	16–32 μs
Repeatability(Z-axis)	0.4 μm
Linearity	±0.1%
Profile data interval(X-axis)	20 μm

**Table 2 sensors-19-00381-t002:** Error analysis of gauge ladder with compensation ratio 0.9948.

Ladder Level	Mean Error	RMS
1.08 mm	0.073 μm	4.963 μm
1.5 mm	4.456 μm	3.424 μm
2 mm	3.665 μm	2.774 μm
**Gauge tolerance**	0.30 μm	
**Accuracy grade**	**2**	

**Table 3 sensors-19-00381-t003:** The results of inverse calculation error by correcting 6 angles of view.

Deg (°)	$O_{n}\;(\mathrm{mm})$	$O_{n}^{\prime }\;(\mathrm{mm})$	$\Delta x_{n}\;(\mathrm{mm})$	RMS (mm)
0	(4.983, 3.425, −11.961)	(4.983, 3.425, −11.961)	0	(0.014, 0.033, 0.021)
60	(9.061, 3.433, −13.718)	(4.994, 3.420, −11.969)	0.014	(0.009, 0.077, 0.027)
120	(12.644, 3.393, −11.062)	(4.985, 3.410, −12.004)	0.046	(0.013, 0.048, 0.011)
180	(12.125, 3.377, −6.674)	(4.960, 3.437, −11.970)	0.028	(0.011, 0.085, 0.024)
240	(7.999, 3.347, −4.947)	(5.025, 3.421, −11.980)	0.046	(0.008, 0.026, 0.024)
300	(4.442, 3.347, −7.570)	(4.974, 3.390, −11.997)	0.051	(0.006, 0.016, 0.026)

**Table 4 sensors-19-00381-t004:** The turntable calibration and fitting statistics of the spherical surface.

Fitting Result		Calibration Evaluation	
Parameter	Value	Parameter	Value
Fitted $R_{B}^{\prime }$ (mm)	9.986	Axis center $O^{\prime }$ (mm)	(8.542, 3.387, −9.322)
Total points number	80,597	Shaft angle of axis (°)	0.713
Average error (mm)	±0.004	Diameter (mm)	20
Maximum error (mm)	0.229	Diameter accuracy (μm)	0.25
RMS σ (mm)	0.012	Roundness (μm)	0.3

**Table 5 sensors-19-00381-t005:** The statistical results of the two methods.

Measurement Number	Geomagic Measurement Results			Measurement Results in this Paper		
	x	y	z	x	y	z
1	5.7094	8.0886	−14.2365	5.7088	8.0822	−14.2381
2	5.7056	8.0377	−14.2013	5.7058	8.0385	−14.2014
3	5.6926	8.0184	−14.1902	5.6934	8.0207	−14.1904
4	5.7042	8.0722	−14.2083	5.6958	8.0718	−14.2054
5	5.6992	8.0408	−14.1931	5.7025	8.0453	−14.1967
6	5.7065	8.0188	−14.2085	5.7083	8.0219	−14.2099
7	5.6922	8.0464	−14.2019	5.6924	8.0467	−14.2022
8	5.7018	8.0692	−14.2415	5.7051	8.0711	−14.2430
9	5.6876	8.0365	−14.2100	5.6871	8.0361	−14.2084
10	5.7039	8.0296	−14.2317	5.7045	8.0274	−14.2334
11	5.6822	8.0218	−14.1976	5.6818	8.0245	−14.1983
12	5.6928	8.0367	−14.2145	5.6925	8.0306	−14.2172

**Table 6 sensors-19-00381-t006:** The statistical results of the two methods.

Dimension	Value
The average number of MVS point cloud	49,078
Time spent in each view (s)	8.5
Average error (mm)	0.075
RMS (mm)	0.151
RMS (%)	0.604

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
