# Peer review of "A Multi-View Stereo Measurement System Based on a Laser Scanner for Fine Workpieces"

_sensors, 2019, doi:10.3390/s19020381_

Round 1
Reviewer 1 Report
1. Is equation (10) correct? Please check it.
2. What is the “MAE”? Please provide the full text.
3. …until the change of δ is less than a threshold. How to define this threshold?

Author Response
Point 1: Equation (10) f(??′,??′,??′)=Σ(√(??′−??)2+(??′−??)2+(??′−??)2??=1−?? = min,
Is correct? Please check it.
Response 1: Thanks for your comment. The correct equation has been changed to,
(Line 245)
Point 2: What is the “MAE”? Please provide the full text.
Response 2: Thank you for the comments. MAE means Mean Absolute Deviation, I added the full text you mentioned in the article (Line 286)
Point 3: until the change of δ is less than a threshold. How to define this threshold?
Response 3: If the method does not converge to a minimum or the convergence rate is too slow, a threshold must be set to stop the iteration, which can be set according to the system error, I set it to 1e-4 in program (Line 261)

Reviewer 2 Report
The article concerns a very interesting topic of multi-view stereo measurement system which Authors based on laser scanner. The paper is well structured and presents the methodology, used hardware and appropriately presents results.
General remarks:
1. Authors should remove personal pronouns such as ‘we’or ‘our.’
2. I suggest that Figure 12 should be presented in a more detailed way. Could Authors add some markers to facilitate the figure analysis?
3. The paper should undergo layout and language editing. There are many missing spaces between values and units. Some paper parts need appropriate formatting adjusting to the Sensors requirements;
4. Authors should provide a more detailed literature review and compare with proposed in the paper technique.
Detailed remarks:
Line:
14: ‘we’;
19: missing space (‘0.075mm’);
20: missing space (‘34s’);
61: incomplete sentence?
86: ‘3d’;
121: missing space ‘405nm’;
140: Caption of Figure 4 is on the next page;
153: missing space;
313: Table 5 should be moved to the next page;
Author Response
Point 1: Authors should remove personal pronouns such as ‘we’or ‘our.’
Response 1: Thanks for your suggestions. I have checked personal pronouns and More objective expressions were used in this paper. (Line 14, line 35, line 67, line 111, line 137, line182, line 184, )
Point 2: I suggest that Figure 12 should be presented in a more detailed way. Could Authors add some markers to facilitate the figure analysis?
Response 2: Thanks for your suggestions. The structure of the image has been adjusted and necessary markers about point cloud data views has added. (Line 292-295)
Point 3: The paper should undergo layout and language editing. There are many missing spaces between values and units. Some paper parts need appropriate formatting adjusting to the Sensors requirements;
Response 3: Thanks for your comment. The layout and language in this paper has been reviewed and adjusted for MDPI Classes and Styles.
Point 4: Authors should provide a more detailed literature review and compare with proposed in the paper technique.
Response 4: Thank you for the comment. I have made further study of the literatures and provide more detailed literature reviews and comparisons. (Line 50-53, line 56-59, line 64-66, line 79-72, line 47-48)
Line:
14: ‘we’;
19: missing space (‘0.075mm’);
20: missing space (‘34s’);
61: incomplete sentence?
86: ‘3d’;
121: missing space ‘405nm’;
140: Caption of Figure 4 is on the next page;
153: missing space;
313: Table 5 should be moved to the next page;
Response: Thank you for the comments. I have modified the details you remarked and other details to standard format.
